# GSDME in Endothelial Cells: Inducing Vascular Inflammation and Atherosclerosis via Mitochondrial Damage and STING Pathway Activation

**DOI:** 10.3390/biomedicines11092579

**Published:** 2023-09-20

**Authors:** Shiyao Xie, Enyong Su, Xiaoyue Song, Junqiang Xue, Peng Yu, Baoli Zhang, Ming Liu, Hong Jiang

**Affiliations:** 1Department of Cardiology, Shanghai Institute of Cardiovascular Diseases, Zhongshan Hospital, Fudan University, Shanghai 200032, China; 2Innovative Center for New Drug Development of Immune Inflammatory Diseases, Ministry of Education, Fudan University, Shanghai 201203, China; 3National Clinical Research Center for Interventional Medicine, Zhongshan Hospital, Fudan University, Shanghai 200032, China; 4Department of Endocrinology and Metabolism, Fudan Institute of Metabolic Diseases, Zhongshan Hospital, Fudan University, Shanghai 200032, China; 5Department of Health Management Center, Zhongshan Hospital, Fudan University, Shanghai 200032, China; 6Shanghai Engineering Research Center of AI Technology for Cardiopulmonary Diseases, Zhongshan Hospital, Fudan University, Shanghai 200032, China

**Keywords:** GSDME, atherosclerosis, vascular inflammation, endothelial cells, mitochondrial damage, STING pathway

## Abstract

The initiation of atherosclerotic plaque is characterized by endothelial cell inflammation. In light of gasdermin E’s (GSDME) role in pyroptosis and inflammation, this study elucidates its function in atherosclerosis onset. Employing Gsdme- and apolipoprotein E-deficient (*Gsdme^−/−^/ApoE^−/−^*) and *ApoE^−/−^* mice, an atherosclerosis model was created on a Western diet (WD). In vitro examinations with human umbilical vein endothelial cells (HUVECs) included oxidized low-density lipoprotein (ox-LDL) exposure. To explore the downstream mechanisms linked to GSDME, we utilized an agonist targeting the stimulator of the interferon genes (STING) pathway. The results showed significant GSDME activation in *ApoE^−/−^* mice arterial tissues, corresponding with atherogenesis. *Gsdme^−/−^/ApoE^−/−^* mice displayed fewer plaques and decreased vascular inflammation. Meanwhile, GSDME’s presence was confirmed in endothelial cells. GSDME inhibition reduced the endothelial inflammation induced by ox-LDL. GSDME was linked to mitochondrial damage in endothelial cells, leading to an increase in cytoplasmic double-stranded DNA (dsDNA). Notably, STING activation partially offset the effects of GSDME inhibition in both in vivo and in vitro settings. Our findings underscore the pivotal role of GSDME in endothelial cells during atherogenesis and vascular inflammation, highlighting its influence on mitochondrial damage and the STING pathway, suggesting a potential therapeutic target for vascular pathologies.

## 1. Introduction

Atherosclerosis is a prevalent disease that imposes a significant global burden due to its association with ischemia and infarction in vital organs, such as the heart and brain [1,2]. The initial stages of atherosclerotic plaque formation involve the inflammation of endothelial cells, which release adhesion molecules and chemokines, such as intercellular adhesion molecule-1 (ICAM-1), vascular cell adhesion molecule-1 (VCAM-1), and monocyte chemotactic protein-1 (MCP-1). These molecules facilitate the recruitment of circulating monocytes to the intima, where they transform into foam cells within the expanding atherosclerotic plaque [3,4,5].

Gasdermins (GSDMs) are a family of proteins that play a crucial role in pyroptosis, a recently discovered form of programmed cell death characterized by cellular rupture and the release of pro-inflammatory cytokines. In humans, this protein family consists of six paralogous genes: GSDMA, GSDMB, GSDMC, GSDMD, GSDME (also known as DFNA5), and PJVK (also known as DFNB59) [6,7,8,9,10]. Recent studies have revealed the widespread occurrence of pyroptosis in atherosclerosis [11,12], with GSDMD showing a propensity to promote the development of atherosclerosis [13]. Similarly, GSDME, another member of the gasdermin family, induces pyroptosis via the caspase 3-GSDME pathway and is expressed in endothelial cells [14,15,16]. Importantly, caspase 3, an upstream molecule of GSDME, has been identified as an independent risk factor for atherosclerosis in clinical investigations [17].

Aside from its direct involvement in inducing cell pyroptosis, the N-terminal fragment of gasdermin E (GSDME-NT), the activated fragment of GSDME, can aggregate on the mitochondrial membrane, leading to mitochondrial damage and the subsequent release of mitochondrial DNA (mtDNA) into the cytoplasm [18,19]. Recent investigations have revealed that the cyclic GMP-AMP synthase–stimulator of the interferon gene (cGAS-STING) signaling pathway can be activated via excessive cytoplasmic endogenous DNA, including mtDNA or exogenous DNA. This activation results in an increased production of type I interferon [20,21]. Additionally, STING regulates the downstream signaling pathway of nuclear factor kappa-B (NF-κB), which leads to the expression of genes encoding pro-inflammatory cytokines [22]. Recent research suggests that cigarette smoke induces the release of mtDNA from mitochondria in endothelial cells, promoting atherosclerosis via activating the STING pathway. Free mtDNA has emerged as an independent predictor of atherosclerosis risk [17].

In this study, Gsdme- and apolipoprotein E-deficient (*Gsdme^−/−^/ApoE^−/−^*) mice exhibited reduced mitochondrial damage, STING pathway activation, vascular inflammation, and atherosclerotic lesions compared to *ApoE^−/−^* mice after a WD. In in vitro models with human umbilical vein endothelial cells (HUVECs), the inhibition of GSDME was found to ameliorate oxidized low-density lipoprotein (ox-LDL)-induced STING pathway activation and endothelial cell inflammation. Intriguingly, when treated with SR-717, a specific STING agonist, there was a partial reversal of the inflammation and atherosclerosis suppression induced via GSDME inhibition, evident in both in vivo and in vitro settings. Collectively, our findings underscore the pivotal role of endothelial GSDME in atherogenesis, mediated through mitochondrial damage and the subsequent activation of the STING pathway.

## 2. Materials and Methods

### 2.1. Animal Models and Procedures

Animal protocols were approved by the Institutional Animal Care and Use Committee of Shanghai Model Organisms Center, Inc., Shanghai, China (Protocol No. 2020-0045, 31 December 2020). *Gsdme^−/−^* mice and *ApoE^−/−^* mice on the *C57BL/6J* background were obtained from the Shanghai Model Organisms Center, Inc. (Shanghai, China) and crossed to generate *Gsdme^−/−^/ApoE^−/−^* mice. The genotyping of *Gsdme^−/−^* mice was performed by amplifying a 920-bp fragment for the wild-type (WT) allele and a 495-bp fragment for the conventional knockout allele using the forward primer TTGGGGCGGGAAAGGTC and the reverse primer AAGCAGGGCAGTTACAGGAG. Only male mice were included. They were fed a WD (21% fat, 0.15% cholesterol, SLAC, Shanghai, China) from 5 to 18 weeks of age and received intraperitoneal injections of SR-717 (0.3% dimethyl sulfoxide, 10 mg/kg per day; MCE, Monmouth Junction, NJ, USA) or phosphate buffer saline (PBS) in the last 4 weeks. *ApoE^−/−^* mice were sacrificed at 10, 14, or 18 weeks, and all *Gsdme^−/−^/ApoE^−/−^* mice were sacrificed at 18 weeks.

### 2.2. Hematoxylin and Eosin (HE) Staining

Aortic roots from anesthetized mice were fixed in 4% paraformaldehyde (Servicebio, Wuhan, China) for 4 h, washed thrice with PBS, and dehydrated in 30% sucrose overnight. Sections (10 μm thick) were cut, with six per slide, spaced by 80 μm, displaying a three-valve structure. Slides were stored at −80 °C. For HE staining, sections were ethanol dehydrated, cleared in xylene, and mounted with neutral balsam. Images were taken using a Leica microscope (Leica, Wetzlar, Germany). The average atherosclerotic lesion area was quantified from six sections per mouse using ImageJ software version 2 (National Institutes of Health, Bethesda, MD, USA).

### 2.3. Immunofluorescence

OCT-embedded sections were air-dried, treated with 95% ethanol to remove excess OCT, and washed with PBS. They were then permeabilized and blocked using 0.1% Triton X-100, 5% BSA, and 2% donkey serum. The sections were incubated overnight at 4 °C with the following primary antibodies: rat anti-EGF-like module-containing mucin-like hormone receptor-like 1 (F4/80^+^) antibody (1:100, Abcam, Cambridge, UK), rabbit anti-GSDME/DFNA5 antibody (1:50, arigo, Taiwan, China), and rat anti-platelet endothelial cell adhesion molecule (CD31) antibody (1:50, BD Biosciences, Franklin Lakes, NJ, USA). On the next day, sections were exposed to respective secondary antibodies: goat anti-rat IgG (H+L) conjugated with Alexa Fluor 568 secondary antibody (1:400, Abcam, Cambridge, UK), goat anti-rabbit IgG (H+L) with the highly cross-adsorbed secondary antibody, Alexa Fluor 555 (1:500, Invitrogen, Carlsbad, CA, USA), and donkey anti-rat IgG (H+L) with the highly cross-adsorbed secondary antibody, Alexa Fluor 488 (1:500, Invitrogen, Carlsbad, CA, USA). After staining with an antifade mounting medium containing 4′-6-diamidino-2-phenylindole (DAPI) (Beyotime Biotechnology, Beijing, China), fluorescent images were captured using a Leica microscope. F4/80^+^ staining areas were quantified using ImageJ software based on six sections per mouse.

### 2.4. Cell Culture

Human THP-1 monocytes (Cell Bank of Chinese Academy of Sciences, Shanghai, China) were cultured in RPMI 1640 medium (Keygen BioTECH, Nanjing, China) supplemented with 10% fetal bovine serum (FBS; ThermoFisher Scientific, Waltham, MA, USA) and 100 U/mL penicillin plus 100 μg/mL streptomycin (YEASEN, Shanghai, China). HUVECs (National Stem Cell Transformation Resource Center, Shanghai, China) were cultured in the endothelial cell medium (ECM; ScienCell Research Laboratories, Carlsbad, CA, USA) supplemented with 10% FBS, endothelial cell growth supplement (ECGS), and 100 U/mL penicillin plus 100 μg/mL streptomycin. All cells were incubated at 37 °C with 5% CO_2_ in a humidified atmosphere.

### 2.5. Small Interfering RNA (siRNA) Transfection

HUVECs were cultured in six-well plates with 1% FBS ECM supplemented with ECGS without antibiotics. When HUVECs reached 70–80% confluence, lipofectamine 3000 transfection reagent (Invitrogen, Carlsbad, CA, USA) was used to transfect the HUVECs with 100 nmol/L siRNA, namely either GSDME siRNA (si-GSDME) or scramble interfering RNA (si-NC) (Genomeditech, Shanghai, China), for 48 h. After transfection, the cells were cultured with either 100 μg/mL of ox-LDL (Yiyuan Biotechnologies, Guangzhou, China) or PBS for 24 h, with or without 3.6 μmol/L SR-717 or dimethyl sulfoxide (DMSO) added 1 h before and during the exposure to ox-LDL or PBS. The efficiency of the knockout was assessed using Western blotting. The sequences of the siRNAs used were as follows: si-GSDME, 5′-GCGGUCCUAUUUGAUGAUGAA-3′, and si-NC, 5′-UUCUCCGAACGUGUCACGU-3′.

### 2.6. Monocyte–Endothelial Cell Adhesion Assay

HUVECs were exposed to ox-LDL or vehicle for 24 h, with or without SR-717. THP-1 monocytes were labeled with calcein-AM (Beyotime Biotechnology, Beijing, China) and co-cultured with HUVECs for 1 h. After washing to remove non-adherent THP-1 monocytes, cells were fixed, and ImageJ software was used to count cells in 10 random fields.

### 2.7. Western Blot Analysis

Proteins were separated on SDS-polyacrylamide gels (10–12.5%) and transferred to polyvinylidene difluoride membranes (Millipore, Billerica, MA, USA). After blocking with 5% BSA and Tween 20 in Tris-buffered saline, membranes were incubated overnight at 4 °C with the following primary antibodies: rabbit anti-GSDME (1:1000, Abcam, Cambridge, UK), rabbit anti-phosphorylated STING (p-STING; 1:1000, Cell Signaling Technology, Danvers, MA, USA), rabbit anti-STING (1:1000, Cell Signaling Technology, Danvers, MA, USA), rabbit anti-phosphorylated TANK-binding kinase 1 (p-TBK1; 1:1000, Cell Signaling Technology, Danvers, MA, USA), rabbit anti-TBK1 (1:1000, Cell Signaling Technology, Danvers, MA, USA), rabbit anti-phosphorylated interferon regulatory factor-3 (p-IRF3; 1:1000, Cell Signaling Technology, Danvers, MA, USA), mouse anti-IRF3 (1:1000, Santa Cruz, TX, USA), rabbit anti-phosphorylated NF-κB p65 (1:1000, Cell Signaling Technology, Danvers, MA, USA), rabbit anti-NF-κB p65 (1:1000, Cell Signaling Technology, Danvers, MA, USA), rabbit anti-ICAM-1 (1:400, Boster Biological Technology, Wuhan, China), rabbit anti-VCAM-1 (1:400, Boster Biological Technology, Wuhan, China), rabbit anti-MCP-1 (1:400, Boster Biological Technology, Wuhan, China), and β-actin (1:20,000, Bioworld, Nanjing, China). Next, membranes were incubated with either peroxidase AffiniPure goat anti-rabbit IgG (H+L) (1:20,000, Jackson ImmunoResearch, West Grove, PA, USA) or goat anti-mouse IgG (H+L) unconjugated (1:5000, Bioworld, Nanjing, China). Protein bands were visualized using an enhanced chemiluminescence detection system (GE Healthcare Life Sciences, Boston, MA, USA). Band density was quantified using Quantity One software version 4.6.2 (Bio-Rad, Mountain View, CA, USA).

### 2.8. Statistical Analysis

Statistical analyses were performed using GraphPad Prism software version 8.0. Data are presented as mean ± SEM. Normality and equal variance were assessed using the Shapiro–Wilk test and *F* test, respectively. For two-group comparisons, an unpaired Student *t*-test was used, while one-way ANOVA with the Bonferroni post hoc test was employed for multiple group comparisons. Two-way ANOVA with the Bonferroni post-test was used for data with more than two categorical variables. A *p*-value less than 0.05 (*p* < 0.05) was considered statistically significant.

## 3. Results

### 3.1. GSDME Promoted Vascular Inflammation and Atherosclerosis in ApoE^−/−^ Mice

To elucidate the role of GSDME in atherogenesis, *ApoE^−/−^* mice were maintained on a WD regimen, and aortic samples were harvested from these mice aged between 5 and 18 weeks. The Western blot analysis revealed a progressive activation of GSDME in arterial tissues corresponding to the duration of WD (Figure 1A,B), indicating its involvement in atherogenesis.

Therefore, we generated *Gsdme^−/−^/ApoE^−/−^* mice and subsequently validated their genotype via genotyping analysis (Figure 1C). Both *ApoE^−/−^* and *Gsdme^−/−^/ApoE^−/−^* mice were placed on a WD regimen between 5 and 18 weeks of age, and aortic samples were harvested at the 18-week mark. In situ imaging and HE staining provided evidence of a notable reduction in atherosclerotic plaque formation within the aortic root and its branches in *Gsdme^−/−^/ApoE^−/−^* mice compared with *ApoE^−/−^* mice (Figure 1D–F). Additionally, immunofluorescence assays targeting F4/80 revealed a reduced macrophage infiltration in *Gsdme^−/−^/ApoE^−/−^* mice compared to *ApoE^−/−^* counterparts (Figure 1G,H). Consistent with these findings, Western blot assays demonstrated a significant decrease in the protein levels of ICAM-1, VCAM-1, and MCP-1, which are well-established markers of endothelial inflammation, in the aortic tissues of *Gsdme^−/−^/ApoE^−/−^* mice relative to *ApoE^−/−^* mice (Figure 1I–L).

### 3.2. GSDME Mediated Ox-LDL-Induced Inflammation in HUVECs

We utilized immunofluorescence staining techniques to determine the cellular localization of GSDME within aortic tissues. Our analyses demonstrated that GSDME exhibits colocalization with the endothelial cell-specific marker CD31 in the aortas of 18-week-old *ApoE^−/−^* mice (Figure 2A). To delve deeper into the functional implications and underlying mechanisms of GSDME in endothelial cells during atherogenesis, we executed in vitro assays with HUVECs exposed to ox-LDL, thereby mimicking the pathological alterations characteristic of atherosclerosis. By adjusting the concentration of ox-LDL and modulating the exposure duration, we determined that GSDME becomes activated in HUVECs upon ox-LDL exposure. In particular, we noted a significant rise in GSDME-NT levels that directly correlated with both the duration of exposure and the concentrations of ox-LDL (Figure 2B–E).

To investigate the role of GSDME in HUVECs, we employed siRNA specifically targeting GSDME to inhibit its expression. Following 48 h of siRNA treatment, the effectiveness of this knockdown was validated via Western blot analysis, which revealed a marked reduction in GSDME expression with si-GSDME compared to the negative control siRNA (Figure 2F,G). After treatment with either si-GSDME or si-NC, HUVECs were subjected to 100 μg/mL ox-LDL or PBS for 24 h. The si-GSDME treatment significantly attenuated the ox-LDL-induced increase in ICAM-1 and VCAM-1 protein levels, as confirmed via Western blot analyses (Figure 2H–J). Consistent with these observations, monocyte–endothelial cell adhesion assays revealed a significant reduction in THP-1 cell adhesion after si-GSDME treatment in the presence of ox-LDL, relative to the si-NC-treated group (Figure 2K,L).

### 3.3. GSDME Damaged Mitochondria in Endothelial Cells and Activated STING Pathway

Based on the research findings suggesting that GSDME-NT preferentially localizes to the mitochondrial membrane rather than the cell membrane, leading to mitochondrial damage and subsequent translocation of mtDNA (belonging to double-stranded DNA) into the cytoplasm [10,11], we conducted immunofluorescence labeling for cytoplasmic double-stranded DNA (dsDNA) and CD31. This was performed on 5-week-old *ApoE^−/−^* mice (which were not subjected to a WD), 18-week-old *ApoE^−/−^* mice, and 18-week-old *Gsdme^−/−^/ApoE^−/−^* mice. Our findings revealed a marked elevation in dsDNA levels within endothelial cells following WD. However, this rise was notably mitigated in the absence of GSDME, as observed in the *Gsdme^−/−^/ApoE^−/−^* mice (Figure 3A).

Recent investigations have disclosed that cytoplasmic mtDNA, functioning as endogenous DNA, can activate the STING pathway, thereby precipitating an inflammatory response [20]. Concurrently, additional studies have highlighted the activation of the STING pathway within endothelial cells throughout atherogenesis [23,24]. In light of this, we evaluated the activation levels of the STING pathway in the aortas of both *ApoE^−/−^* and *Gsdme^−/−^/ApoE^−/−^* mice at 18 weeks of age following a WD regimen. Our data demonstrated that the phosphorylated activation of critical components, such as STING, TBK1, IRF3, and NF-κB p65, was markedly increased in the *ApoE^−/−^* mice, and this activation was noticeably diminished in the aortic tissues of *Gsdme^−/−^/ApoE^−/−^* mice compared to *ApoE^−/−^* (Figure 3B–F). In vitro findings were consistent with this, as after 24 h of ox-LDL incubation, HUVECs exhibited STING pathway activation, which was mitigated via the intervention of si-GSDME (Figure 3G–K).

### 3.4. Activation of the STING Pathway Altered the Protective Effects of GSDME Deficiency on Atherogenesis and Endothelial Inflammation

In our investigation of the interaction between the STING pathway and GSDME, we utilized SR-717, a STING agonist. The structural and functional analyses confirmed that SR-717 functions as a direct cyclic guanosine monophosphate–adenosine monophosphate (cGAMP) mimetic, inducing the characteristic ‘closed’ conformation of STING [25]. Mice models, both *ApoE^−/−^* and *GSDME^−/−^/ApoE^−/−^*, on a WD regimen, were administered daily intraperitoneal doses of 10 mg/kg SR-717 or a vehicle control from 14 to 18 weeks of age. Utilizing in situ imaging and HE staining, we observed that the activation of the STING pathway led to an enlargement of the plaque area in both *GSDME^−/−^/ApoE^−/−^* and *ApoE^−/−^* mice, thereby counteracting the suppressive influence of GSDME deficiency on atherosclerosis progression (Figure 4A,C). Subsequent analysis employing immunofluorescence methodologies revealed an amplified macrophage infiltration in both mouse models post-STING pathway activation (Figure 4B,D). This intensified inflammatory response served to neutralize the inhibitory effects of GSDME deficiency on vascular inflammation in *ApoE^−/−^* mice. This finding was further corroborated by the altered expression levels of pivotal inflammatory markers, including ICAM-1, VCAM-1, and MCP-1 (Figure 4E–H). Moreover, in the backdrop of GSDME deficiency, a partial restoration in the phosphorylation and activation of the STING pathway was noted upon its activation (Figure 4I–M).

To investigate the influence of the STING pathway on endothelial inflammation mediated via GSDME, HUVECs were initially exposed to siRNA. Following this, they were pre-treated with SR-717 or a vehicle control for a duration of 1 h and subsequently exposed to either ox-LDL or PBS for 24 h. Our results demonstrated that in HUVECs treated with ox-LDL, SR-717 elevated the expression of ICAM-1 and VCAM-1 protein, observed in both the si-NC and si-GSDME groups. Notably, this upregulation counteracted the suppressive influence of si-GSDME on protein expression in the presence of ox-LDL (Figure 5A–C). Simultaneously, our observations revealed that SR-717 enhanced the monocyte–endothelial cell adhesion in response to ox-LDL, counteracting the inhibitory effects of si-GSDME (Figure 5D–E). Additionally, the results from Western blot assays revealed that SR-717 mitigated the reduction in STING pathway phosphorylation, a consequence linked to si-GSDME, in ox-LDL-treated HUVECs (Figure 5F–J).

## 4. Discussion

Our findings provide evidence of GSDME activation during the progression of atherosclerosis and confirm that the deficiency of GSDME exhibited significant effects on reducing atherogenesis and diminishing vascular inflammation in *ApoE^−/−^* mice. Furthermore, we confirmed the localization of GSDME within endothelial cells of arterial tissues. This finding was complemented by in vitro studies, where GSDME deficiency attenuated the inflammatory response in endothelial cells exposed to ox-LDL. Probing further into the mechanistic underpinnings within *ApoE^−/−^* mice, we found that GSDME deficiency correlated with decreased cytoplasmic dsDNA accumulation in endothelial cells, indicating a diminished activation of the STING pathway. Thus, our findings suggest the STING pathway as a potential downstream effector in the cascade of GSDME-induced endothelial inflammation and atherosclerosis. To further elucidate this connection, we employed a STING-specific agonist to activate the STING pathway in GSDME-deficient *ApoE^−/−^* mice and HUVECs. Interestingly, this activation counteracted some of the protective effects of GSDME deficiency, suggesting that the STING pathway operates downstream of GSDME in mediating endothelial inflammation and atherosclerosis.

GSDME, an important member of the gasdermin family responsible for mediating pyroptosis, is involved in multiple inflammatory diseases [26,27,28]. Numerous studies have demonstrated that pyroptosis contributes to inflammation in the progression of atherosclerosis [29,30,31]. However, the role and mechanism of GSDME in atherogenesis remain unclear. This study presents compelling evidence for GSDME’s role in endothelial cells, contributing to vascular inflammation and the onset of atherosclerosis, as demonstrated both in animal models and at the cellular level. In a recent in vitro study using macrophages, it was found that the ablation of GSDME suppressed inflammation and macrophage pyroptosis induced by ox-LDL [32]. However, our findings bring to light a novel aspect, revealing that GSDME mediates endothelial inflammation, thereby promoting vascular inflammation and the initiation of atherosclerosis. This underscores the importance of endothelial cells when considering GSDME’s influence on atherosclerosis.

While prior research has posited the involvement of endothelial cell pyroptosis in atherosclerotic progression, it is important to note that the occurrence rate of this pyroptosis remains relatively low [33]. Given this context, our study was geared toward identifying alternative mechanisms by which GSDME exacerbates vascular inflammation and atherosclerosis beyond the scope of cell pyroptosis. Our study observed elevated cytoplasmic dsDNA levels in *ApoE^−/−^* mice, a trend that significantly receded in the absence of GSDME. This is consistent with earlier studies suggesting that GSDME-NT tends to aggregate on mitochondrial membranes, thereby triggering mitochondrial dysfunction and resulting in the release of materials, such as dsDNA, into the cytoplasm [19]. Importantly, our results showed that inhibiting GSDME reduced the activation of the STING pathway in both vascular tissues and endothelial cells. Consequently, we explored the potential role of SR-717, a known STING agonist [25]. Experiments conducted on mice and HUVECs demonstrated that SR-717 administration could partially counteract the inhibitory effects of GSDME deficiency on atherosclerosis, vascular inflammation, and ox-LDL-induced endothelial cell inflammation. This implies that the STING pathway may function downstream of GSDME-mediated effects. To our knowledge, this is the inaugural study to propose the mtDNA-STING axis as a potential downstream mechanism influenced by GSDME during atherogenesis and endothelial inflammation.

Moreover, our findings indicate that STING agonists exacerbate atherosclerosis, vascular inflammation, and ox-LDL-induced endothelial cell inflammation in *ApoE^−/−^* mice. This observation is particularly intriguing given the recent surge of interest in STING agonists. Numerous studies have highlighted the potential therapeutic benefits of STING agonists across various clinical settings, with a particular focus on their promising role in cancer therapy [34,35,36]. However, our results underscore the need for caution. While the therapeutic potential of STING agonists is undeniable, it is imperative to consider their potential side effects, especially in the context of vascular health. Future studies should delve deeper into the mechanisms underlying these observations to ensure the safe and effective application of STING agonists in clinical settings.

In this investigation, we employed *ApoE^−/−^* and *GSDME^−/−^/ApoE^−/−^* mouse models. For subsequent studies, it may be beneficial to use endothelial-specific Gsdme-deficient mice to delve deeper into GSDME’s role in atherogenesis. Our findings suggest that GSDME activity modulates the mtDNA-STING pathway. However, it is crucial to consider that other signaling cascades may also be influenced downstream of GSDME during atherogenesis. The field has seen recent advancements, with the identification of alternative innate immune sensors for cytosolic mtDNA. In a noteworthy study, ZBP1 was identified to potentiate IFN-I responses to cytosolic mtDNA and, in turn, to exacerbate inflammation [37]. Given these insights, future research should focus on uncovering additional potential pathways, aiming to deepen our understanding of the intricate mechanisms through which GSDME affects endothelial inflammation and atherosclerosis. As we continue to unravel these mechanisms, the urgency to develop and validate GSDME protein inhibitors in mouse atherosclerosis models becomes evident. Such advancements are pivotal for enhancing our mechanistic insights and further pave the way for potential clinical drug development.

## 5. Conclusions

This research underscores the pivotal role of GSDME, an often overlooked member of the gasdermin family, in the pathogenesis of atherosclerosis, specifically in endothelial cells. Through both in vivo and in vitro analyses, we have unearthed GSDME’s contribution to vascular inflammation, a key facet of atherogenesis. Our findings underscore the significant effects of GSDME deficiency on reducing atherogenesis and pinpoint the STING pathway as a downstream mediator of these processes. The compelling evidence presented here broadens our understanding of endothelial inflammation within the context of atherosclerosis, highlighting the nuanced interplay between GSDME, mitochondrial dysfunction, and the STING pathway. Importantly, this work opens new avenues for therapeutic strategies targeting GSDME-mediated mechanisms, with potential implications in the prevention and treatment of atherosclerosis.

## Figures and Tables

**Figure 1 biomedicines-11-02579-f001:**
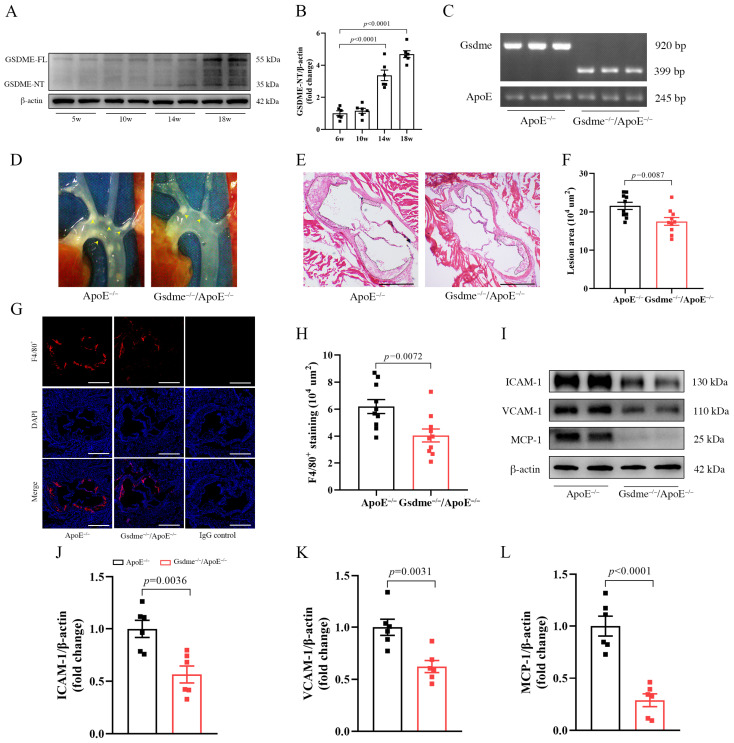
GSDME (gasdermin E) promotes vascular inflammation and atherosclerosis in *ApoE^−/−^* (apolipoprotein E-deficient) mice. (**A**) Representative image of Western blot analysis of GSDME-FL (full-length gasdermin E) and GSDME-NT (N-terminal fragment of gasdermin E) protein levels in aortas of *ApoE^−/−^* mice with varying durations of WD (Western diet) starting from 5-week-old; (**B**) the quantification of GSDME-NT, n = 6; (**C**) the genotyping results of *ApoE^−/−^* mice and *Gsdme^−/−^/ApoE^−/−^* (Gsdme- and apolipoprotein E-deficient) mice for the Gsdme gene and ApoE gene; (**D**–**L**) the *ApoE^−/−^* mice and *Gsdme^−/−^/ApoE^−/−^* mice fed with WD starting at 5 weeks of age underwent subsequent aortic examination at the age of 18 weeks; (**D**) representative images of the root of the murine aorta in situ; (**E**) representative images stained with HE, and scale bars indicate 500 μm. (**F**) Quantitative statistical graph of HE (hematoxylin and eosin) staining, n = 10; (**G**) immunofluorescence staining of EGF-like module-containing mucin-like hormone receptor-like 1 (F4/80^+^, red) and 4′-6-diamidino-2-phenylindole (DAPI, blue), scale bars indicate 500 µm; (**H**) quantitative statistical graph of immunofluorescence staining of F4/80^+^, n = 10; (**I**) representative image of Western blot analysis of ICAM-1 (intercellular adhesion molecule-1), VCAM-1 (vascular cell adhesion molecule-1), MCP-1 (monocyte chemotactic protein-1), and β-actin; (**J**–**L**) the quantification of VCAM-1, ICAM-1, and MCP-1, respectively, n = 6.

**Figure 2 biomedicines-11-02579-f002:**
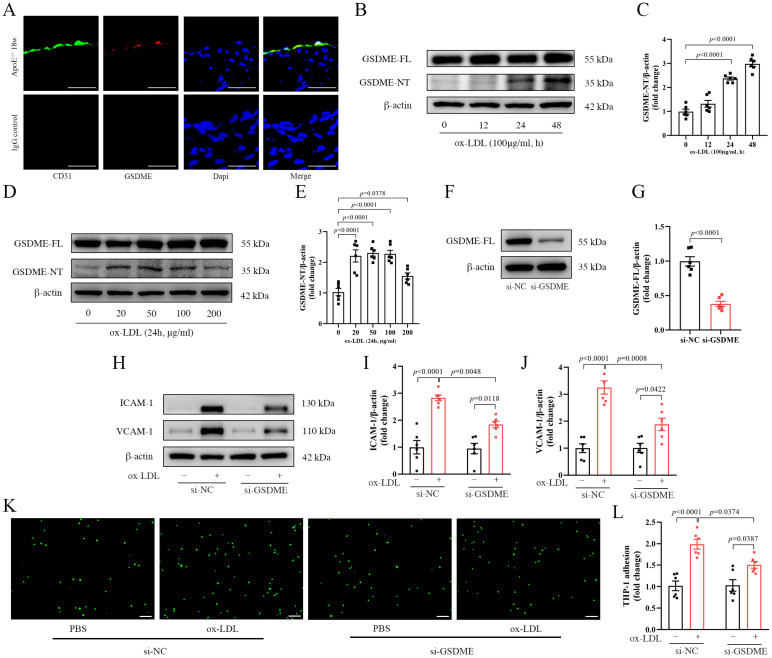
GSDME (gasdermin E) mediated inflammation in ox-LDL(oxidized low-density lipoprotein)-induced HUVECs (human umbilical vein endothelial cells). (**A**) Immunofluorescence staining of platelet endothelial cell adhesion molecule (CD31; green), GSDME (red), and 4′-6-diamidino-2-phenylindole (DAPI; blue) on the aortas of 18-week-old *ApoE^−/−^* (apolipoprotein E-deficient) mice, and scale bars indicate 20 μm; (**B**) representative image of Western blot analysis of GSDME-FL (full-length gasdermin E) and GSDME-NT (N-terminal fragment of gasdermin E) protein levels in HUVECs under varying durations of 100 μg/mL ox-LDL induction and (**C**) the quantification of GSDME-NT normalized to β-actin, n = 6; (**D**) representative image obtained from Western blot analysis of the protein expression levels of GSDME-FL and GSDME-NT in HUVECs exposed to varying concentrations of ox-LDL induction for 24 h and (**E**) the quantification of GSDME-NT normalized to β-actin, n = 6; (**F**) representative Western blot showing the efficiency of siRNA (small interfering RNA)-mediated GSDME knockdown in HUVECs following a 48 h treatment, and (**G**) quantitative statistical analysis of GSDME-FL normalized to β-actin, n = 6; (**H**–**L**) HUVECs were transfected with either si-NC (scramble interfering RNA) or si-GSDME (GSDME siRNA) for a duration of 48 h, followed by exposure to 100 μg/mL of ox-LDL or PBS (phosphate buffer saline) for an additional 24 h; (**H**) representative image from Western blot analysis displaying the expression levels of adhesion molecules ICAM-1 (intercellular adhesion molecule-1), VCAM-1 (vascular cell adhesion molecule-1), and β-actin, and scale bars represent 150 µm; (**I**,**J**) quantification of VCAM-1 and ICAM-1 normalized to β-actin, respectively, n = 6; (**K**) representative image displaying THP-1 cells (green) adhering to HUVECs, scale bars indicate 150 μm; (**L**) quantitative analysis of the cell count of THP-1 cells adhered to HUVECs, n = 6.

**Figure 3 biomedicines-11-02579-f003:**
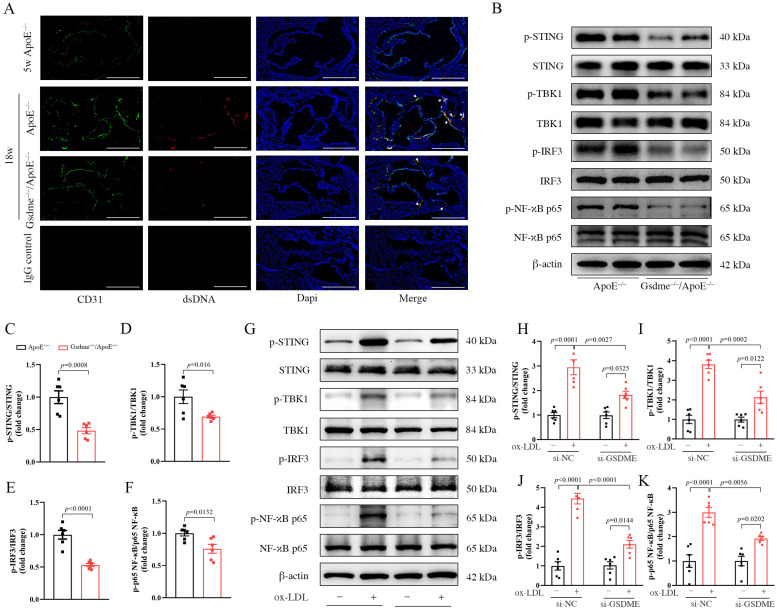
GSDME (gasdermin E) induces mitochondrial damage in endothelial cells and activates the STING (stimulator of the interferon genes) pathway. (**A**–**F**) From 5 weeks of age, mice were placed on a WD (Western diet) regimen. Aortic samples were harvested from *ApoE^−/−^* (apolipoprotein E-deficient) and *Gsdme^−/−^/ApoE^−/−^* (Gsdme- and apolipoprotein E-deficient) mice at 18 weeks of age, as well as from 5-week-old *ApoE^−/−^* mice (not subjected to the WD). (**A**) Immunofluorescence staining of platelet endothelial cell adhesion molecule (CD31; green), double-stranded DNA (dsDNA; red), and 4′-6-diamidino-2-phenylindole (DAPI, blue), scale bars indicate 500 μm; (**B**) representative image of Western blot analysis of p-STING (phosphorylated STING), STING, p-TBK1 (phosphorylated TANK-binding kinase 1), TBK1, p-IRF3 (phosphorylated interferon regulatory factor-3), IRF3, p-NF-κB p65 (phosphorylated nuclear factor kappa-B p65), NF-κB p65, and β-actin; (**C**–**F**) the relative quantification of p-STING to STING, p-TBK1 to TBK1, p-IRF3 to IRF3, and p-NF-κB to NF-κB, n = 6; (**G**–**K**) HUVECs underwent transfection with either si-NC (scramble interfering RNA) or si-GSDME (GSDME siRNA) for a duration of 48 h. Subsequently, these cells were either treated with 100 μg/mL of ox-LDL (oxidized low-density lipoprotein) or exposed to PBS (phosphate buffer saline) for an additional 24 h. (**G**) Representative image of Western blot analysis of p-STING, STING, p-TBK1, TBK1, p-IRF3, IRF3, p-NF-κB p65, NF-κB p65, and β-actin. (**H**–**K**) Relative quantification of p-STING to STING, p-TBK1 to TBK1, p-IRF3 to IRF3, and p-NF-κB p65 to NF-κB p65, respectively, n = 6.

**Figure 4 biomedicines-11-02579-f004:**
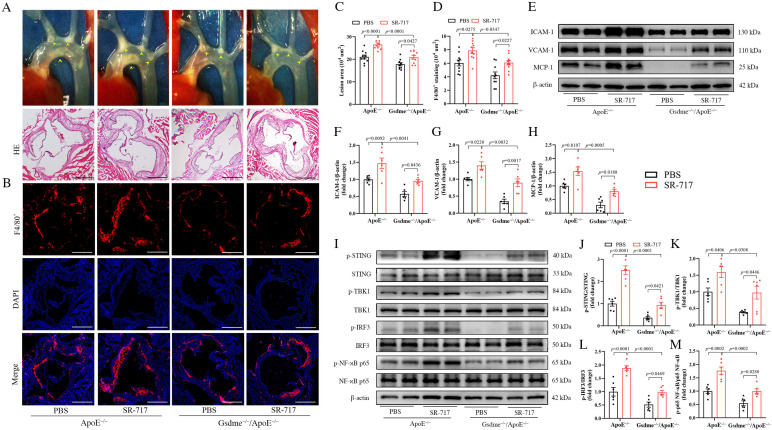
Activation of the STING (stimulator of the interferon genes) pathway partially reversed the inhibitory effects of GSDME (gasdermin E) deficiency on vascular inflammation and atherogenesis. Beginning at 5 weeks of age, *ApoE^−/−^* (apolipoprotein E-deficient) and *Gsdme^−/−^/ApoE^−/−^* (Gsdme- and apolipoprotein E-deficient) mice were administered a WD (Western diet). Starting at 14 weeks, these mice were injected intraperitoneally with either SR-717 or PBS (phosphate buffer saline) at a dosage of 10 mg/kg daily until they reached 18 weeks of age, at which point aortic examinations were performed; (**A**) representative images display the murine aortic root in situ, and representative histological sections stained with HE (hematoxylin and eosin), scale bars indicating 500 µm; (**B**) representative images of immunofluorescence staining for EGF-like module-containing mucin-like hormone receptor-like 1 (F4/80^+^, red) and 4′-6-diamidino-2-phenylindole (DAPI, blue), scale bars indicating 500 µm; (**C**) quantitative statistical analysis of HE staining, n = 10; (**D**) quantitative statistical analysis of F4/80^+^ immunofluorescence staining, n = 10. (**E**) Representative Western blot image shows adhesion molecule expression levels for ICAM-1 (intercellular adhesion molecule-1), VCAM-1 (vascular cell adhesion molecule-1), chemokine MCP-1 (monocyte chemotactic protein-1), and β-actin; (**F**–**H**) quantification of VCAM-1, ICAM-1, and MCP-1 normalized to β-actin, respectively, n = 6; (**I**) representative image of the Western blot analysis of phosphorylated and total levels of STING, TBK1 (TANK-binding kinase 1), IRF3 (interferon regulatory factor-3), NF-κB p65 (nuclear factor kappa-B p65)), and β-actin, with (**J**–**M**) relative quantifications of phosphorylated-to-total ratios for STING, TBK1, IRF3, and NF-κB p65, n = 6.

**Figure 5 biomedicines-11-02579-f005:**
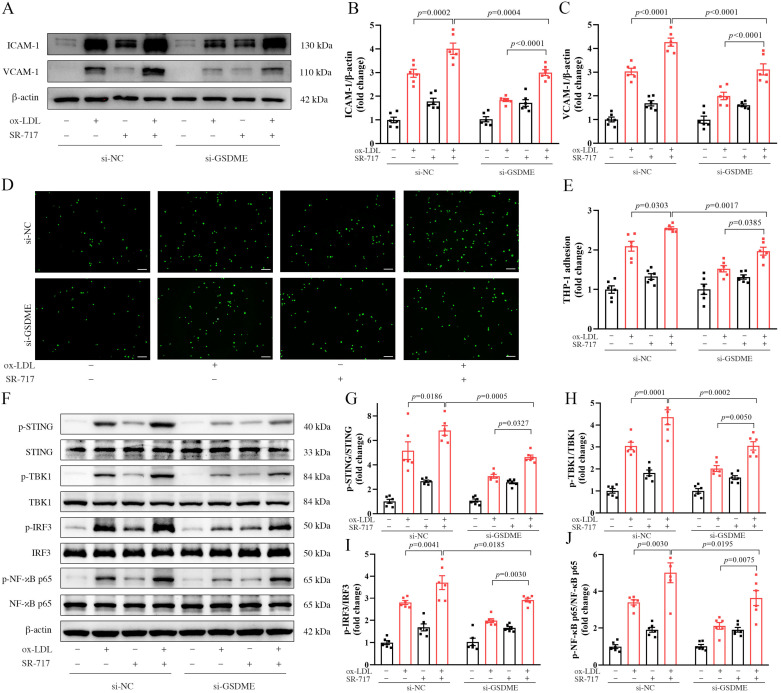
Activation of the STING (stimulator of the interferon genes) pathway mitigated the inhibitory effects of GSDME (gasdermin E) knockdown on ox-LDL(oxidized low-density lipoprotein)-induced endothelial inflammation. HUVECs (human umbilical vein endothelial cells), post-transfection with either si-NC (scramble interfering RNA) or si-GSDME (GSDME siRNA), underwent a pretreatment of 3.6 μmol/L SR-717 or DMSO (dimethyl sulfoxide) for 1 h. These cells were then exposed to 100 μg/mL of ox-LDL or PBS (phosphate buffer saline) for a duration of 24 h before proceeding with the subsequent assays. (**A**) Representative Western blot image of adhesion molecules ICAM-1 (intercellular adhesion molecule-1), VCAM-1 (vascular cell adhesion molecule-1), and β-actin; (**B**,**C**) quantification of VCAM-1 and ICAM-1, normalized to β-actin, respectively, n = 6; (**D**) representative image of THP-1 cells (green) adhering to HUVECs, scale bars indicate 150 µm; (**E**) quantitative statistical assessment of adhered THP-1 cell counts, n = 6; (**F**) representative Western blot image of phosphorylated and total levels of STING, TBK1 (TANK-binding kinase 1), IRF3 (interferon regulatory factor-3), NF-κB p65 (nuclear factor kappa-B p65)), and β-actin; (**G**–**J**) relative quantifications of the phosphorylated forms to their corresponding total proteins, n = 6.

## Data Availability

Data supporting the findings of this study are available within the article. Further data can be obtained upon reasonable request from the corresponding author.

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
