# Peer review of "GSDME in Endothelial Cells: Inducing Vascular Inflammation and Atherosclerosis via Mitochondrial Damage and STING Pathway Activation"

_biomedicines, 2023, doi:10.3390/biomedicines11092579_

Round 1
Reviewer 1 Report
The present study demonstrated that GSDME could Induce vascular inflammation followed by atherosclerosis via mitochondrial damage and STING Pathway in endothelial cells by both in vitro and in vivo study. A reasonable animal model with genetic modification was used. The novelty and clinical relevance of this study is adequate and the methodology is reasonable. However, there are still some minor issues needed to be addressed.
1. There should be somehow marked or described in the representative HE staining (figure 1E) to make reader easy to recognize how the author quantified in figure 1F.
2. The description of scale bar in figure 2K should be added.
3. The difference of molecular weight between total and phosphorylation form of both STING and IRF3 is unreasonable.
4. The representative photo in figure 4A did not match the description in the main text.
5. In Figure 5, the label for each group did not match the description in the legend.
Minor editing of English language required
Reviewer 2 Report
This is an interesting study, however, there are several concerns needing to be addressed for strengthening the impact of this study.
1. The title style of references was not consistent; please revise the section of references.
2. For the convenience of readers, the graphic abstract is highly recommended.
3. What is the distribution of key molecules involved in the STING pathways (STING, TBK1, IRF3 and NF-kB) in atherosclerotic lesion? Immunohistochemistry for the molecules is required for this review point.
Reviewer 3 Report
This is a basic study, which underscored the pivotal role of GSDME in endothelial cells during atherogenesis and vascular inflammation, highlighting its influence on mitochondrial damage and the STING pathway, suggesting a potential therapeutic target for vascular pathologies.
This reviewer considers that the authors well performed the present study. This reviewer has some comments as described below.
Major comments:
1. Figures were too small to see. The authors should make them bigger.
2. The authors should discuss about future perspective in the Discussion section.
Author Response
请参阅附件。

Round 2
Reviewer 2 Report
Authors have adequately addressed my comments.
Reviewer 3 Report
This reviewer has no further comment.